

# A new species of *Scapholeberis* Schoedler, 1858 (Anomopoda: Daphniidae: Scapholeberinae) from the Colombian Amazon basin highlighted by DNA barcodes and morphology

Camilo Andrade-Sossa[1], Lorena Buitron-Caicedo[1] and Manuel Elías-Gutiérrez[2]

[1] Grupo de Estudios en Recursos Hidrobiológicos Continentales. Departamento de Biología, Universidad del Cauca, Popayán, Colombia
[2] Departamento de Sistemática y Ecología Acuática, El Colegio de la Frontera Sur, Chetumal, Quintana Roo, México

Corresponding author
Manuel Elías-Gutiérrez, melias@ecosur.mx

## ABSTRACT

**Background:** The Amazon basin is recognized as one of the most complex and species-rich freshwater environments globally. The diversity of zooplankton here remains unknown, with many species undescribed.

**Methods:** Here, we describe a new species of *Scapholeberis* Schoedler, 1858 (Cladocera: Anomopoda: Daphniidae: Scapholeberinae) from the Colombian Amazon Basin, collected with recently designed light traps. The description is based on detailed morphology (based on SEM and light microscopy) of parthenogenetic females, ephippial females, males, and molecular data based on the COI gene.

**Results:** *Scapholeberis yahuarcaquensis* n. sp. has a combination of characters present in *Scapholeberis kingi* Sars, 1888 and *Scapholeberis armata freyi* Dumont & Pensaert, 1983. These are a trilobate rostrum, with the middle lobe well developed with sides straight to relatively rounded, the presence of an elongated slit frontal head pore, a dorsal pore in the juncture of the cephalic shield and the valves, and a single denticulate membrane at the posterior rim of the valves, with stronger setae in the last third. The unique characters of the parthenogenetic females are ventral sucker with delicate triangles. Each has a filament-like projection in the lamellae's inner side and an external section forming convex folds with denticle-like projections in the middle zone of the sucker-plate. There is a peculiar pitted sculpture in the ephippial females and a strong projection in the front of it. The male hook on the limb I with a blunt tip, a quirky lamella-like outgrow in the proximal side, and a paddle with well-developed spines scattered on its surface. The ventral sucker-lamellae in the male is much more developed than the female. The COI gene sequences showed an interspecific mean genetic divergence of 16.4% between *S. yahuarcaquensis* n. sp. and the closest species *S. freyi* from Mexico, supporting our results. A coalescence analysis and Barcode Index Number also support the new species based on the DNA sequences. New methods of collecting and integrative

biology will give important support to recognize the fauna from the Amazon Basin, one of the most important sources of fresh water in the world that remains unknown in many respects.

## INTRODUCTION

*Scapholeberis* Schoedler, 1858 is one of the two genera of the Scapholeberinae *Dumont & Pensaert, 1983* included in the Daphniidae Straus, 1820. They are small cladocerans adapted to live in the hyponeuston of the freshwater ecosystems. A peculiar adaptation of this genus is the ventral sucker to attach the animal by superficial tension below the water surface. This structure was described by the first time in detail with scanning electron photographs by *Dumont & Pensaert (1983)* for different members of the Scapholeberinae.

It seems that these daphniids have a long history living suspended below the surface film of the water, with the oldest *Scapholeberis*-like member found in amber dated from the Lower Cretaceous (c. 100 My before present) (*Flössner & Fryer, 2016*). However the affinity of this specimen was doubted by *Van Damme & Kotov (2016)* without analyzing the original material. Out of discussions, if we consider a long history of the genus, it is unlikely to have only eight species known worldwide. For example, *Quiroz-Vazquez & Elías-Gutiérrez (2009)* highlighted the slow advance in the taxonomic knowledge of *Scapholeberis*. They described the last member of this genus from Mexico by using integrative taxonomy. It was related to *Scapholeberis armata freyi Dumont & Pensaert, 1983*, now recognized as *Scapholeberis freyi* by *Kotov et al. (2013)* and *Taylor, Connelly & Kotov (2020)*.

Recently *Taylor, Connelly & Kotov (2020)* published a phylogeographic analysis of Scapholeberinae based in three RNA regions of the mitochondrial genome showing 17 divergent lineages across the world. Nevertheless, South American fauna was poorly represented in their dataset. In the particular case of the Amazon basin, no barcoding study exists related to zooplankton. However, it is a unique enormous eco-region with many endemic species (*Rowe, Adamowicz & Hebert, 2007*). Currently, while we were preparing a baseline for several lakes and rivers from the Colombian Amazon Basin, we found an intriguing *Scapholeberis* with some resemblance to *Scapholeberis freyi* (*Dumont & Pensaert, 1983*). In the checklists and data published from South America there are five members of this genus (*De los Rios-Escalante & Kotov, 2015*; *Kotov & Fuentes-Reines, 2015*), one of them identified as *Scapholeberis armata freyi* in Brazil (*De Abreu, 2016*; *Elmoor-Loureiro, 2000*), and represented by the sequence KU315490.1 in GenBank.

This work aims to describe a new species based on all possible morphological characters, and molecular data, based on COI sequences and a comparison of them with other species from the world.

## MATERIALS AND METHODS

On May 2 and 3 of 2018, we deployed a light trap and carried out several horizontal tows with a plankton net of 50 μm mesh in several locations along the Amazonas basin from Colombia. The light trap was the same as described by *Montes-Ortiz & Elías-Gutiérrez (2018)*, and it worked only 3–4 h to avoid oversaturation and immediate death of the collected material. We noticed that the species from this region have a strong positive phototaxis. Specimens were preserved according to *Elías-Gutiérrez et al. (2018)*, and all of them were sorted under a stereomicroscope. Part of the material was processed for molecular analyses, and other specimens were used for a detailed morphological study.

Field permit for collections with non-commercial scientific research purposes was issued by Autoridad Nacional de Licencias Ambientales (ANLA) (Resolution #0152).

### Morphological analysis

For morphology analyses, we mounted the dissections in a mixture of glycerol and formaldehyde (1:1) on a slide and covered them with a coverslip. The slide was sealed with Depex mounting medium. We analyzed all specimens under a microscope with phase contrast and Normanski microscopy (BX51; Olympus, Tokyo, Japan). All limbs were drawn using a camera lucida attached to the microscope and digitized directly with a CINTIQ tablet (WACOM, Saitama, Japan) using Adobe Photoshop PS4.

Part of the material was prepared using the drying method with Hexamethyldisilazane (HDMS), to avoid deforming of the delicate membranes of the ventral sucker. They were passed in gradual concentrations, increasing 10% each time, diluted with 100% ethanol, from 30% to 100%. This work was done after dehydration of the specimens to absolute ethanol. Finally, after evaporation of the HDMS, in the final step, they were gold coated. All photographs and observations were made with a scanning electron microscope (SEM) JEOL-JSM6010 at the Chetumal Unit from El Colegio de la Frontera Sur, Mexico.

### DNA extraction and amplification

DNA was extracted, and thermocycling was performed accordingly to the methods provided by *Elías-Gutiérrez et al. (2018)*, with the primers developed by *Prosser, Martínez-Arce & Elías-Gutiérrez (2013)*. PCR products were visualized on a 2% agarose gel using an E-Gel 95 well Pre-cast Agarose Electrophoresis System (Invitrogen, Carlsbad, CA, USA) and those showing a band, as a result of the PCR product, were selected for sequencing.

### Sequencing and data analysis

PCR products were sequenced bi-directionally with the methods suggested by *Hajibabaei et al. (2005)*, using M13F and M13R primers. Sequences were edited accordingly with the methods provided by *Elías-Gutiérrez et al. (2018)*. Sequence data, electropherograms, trace files, primer details, photographs, and collection localities for all specimens are available within the dataset *Scapholeberis* from Amazon Basin (DS-SCAMACOL) in Barcode of Life Database (BOLD, www.boldsystems.org). All data were analyzed with the tools provided on BOLD for control of quality (*Ratnasingham & Hebert, 2007*).

In all cases, the vouchers were deposited at the Museo de Historia Natural, Instituto de Ciencias Naturales, Universidad Nacional de Colombia, and the Reference Collection at El Colegio de la Frontera Sur, Unidad Chetumal (Access numbers ECO-CH-Z 10336-10338).

With all sequences from the Amazon, we prepared a dataset with the name DS-SCAMACOL *Scapholeberis* from Amazon Basin in BOLD database (DOI: 10.5883/DS-SCAMACOL). Additionally, these sequences were uploaded to GenBank (Accession numbers: MT607962–MT607970).

We used the Alignment Transformation Environment (ALTER, http://www.sing-group.org/ALTER/), to get representative haplotypes of all the genetic variants of COI within and between species (Supplemental File 1). We applied MrBayes Vers. 3.2.5 and jModelTest 2.1.10 for calculations of the phylogeny. The best fit substitution model was TIM2+G, and it was replaced by GTR+G in MrBayes along with 1 M generations to obtain a tree. The Generalized Mixed Yule Coalescent (GMYC) method was applied as *Gutierrez-Aguirre et al. (2020)* suggested to delimit the species. Finally, with the groups delimited by the GMYC technique, we calculated the mean estimates of evolutionary divergence over sequence pairs between groups. Analyses were conducted using the Kimura 2-parameter model (*Kimura, 1980*) in MEGA 7 (*Kumar, Stecher & Tamura, 2016*). The standard error of the estimates was obtained by a bootstrap procedure (500 replicates).

## Nomenclatural acts

The electronic version of this article in Portable Document Format will represent a published work according to the International Commission on Zoological Nomenclature (ICZN), and hence the new names contained in the electronic version are effectively published under that Code from the electronic edition alone. This published work and the nomenclatural acts it contains have been registered in ZooBank, the online registration system for the ICZN. The ZooBank Life Science Identifiers (LSIDs) can be resolved and the associated information viewed through any standard web browser by appending the LSID to the prefix http://zoobank.org/. The LSID for this publication is: urn:lsid:zoobank.org:pub:5CF5417E-A35F-42BC-97D4-5BB4AD6D6259. The online version of this work is archived and available from the following digital repositories: PeerJ, PubMed Central and CLOCKSS.

## RESULTS

## SYSTEMATICS

Class Branchiopoda Calman, 1909
Order Anomopoda Sars, 1865
Family Daphniidae Straus, 1820
Subfamily Scapholeberinae *Dumont & Pensaert, 1983*
Genus *Scapholeberis* Schoedler, 1858
*Scapholeberis yahuarcaquensis* n. sp.

**Type Locality:** Quebrada and lakes system Yahuarcaca, Amazon basin. A system of interconnected blackwater streams and lakes, with periodic floodings of white-water from the Amazon River. Three localities in this interconnected system, located at 4.193640° S, 69.954465° W (Yahuarcaca Lake); 4.158175° S, 69.968207° W (Quebrada Yahuarcaca, Flooded Trees) and 4.157807° S, 69.967763° W (Quebrada Yahuarcaca, near village San Antonio) were positive for this species. From the three, we designate the last as the type locality (Quebrada Yahuarcaca near the village San Antonio).

**Etymology:** This species is named after its *terra typica*, the Yahuarcaca lakes and stream system in the Amazon basin from Colombia.

**Holotype:** Adult parthenogenetic female in 96% ethanol with addition of a drop of glycerol, deposited at the Museo de Historia Natural, Instituto de Ciencias Naturales, Universidad Nacional de Colombia. Access number, ICN-MHN-CR 3405.

**Allotype:** One complete male in 96% ethanol with addition of a drop of glycerol, deposited at the Museo de Historia Natural, Instituto de Ciencias Naturales, Universidad Nacional de Colombia Access number, ICN-MHN-CR 3406

**Paratypes:** Five parthenogenetic females and one ephippial female in 96% ethanol with addition of a drop of glycerol, deposited at the Museo de Historia Natural, Instituto de Ciencias Naturales, Universidad Nacional de Colombia Access numbers, ICN-MHN-CR 3407 and ICN-MHN-CR 3408. Four parthenogenetic and two ephippial females in 96% ethanol with addition of a drop of glycerol, one dissected parthenogenetic female in a mix formalin-glycerol, sealed with GURR mounting medium, deposited at El Colegio de la Frontera Sur, Unidad Chetumal, Access numbers ECO-CH-ZPL10336-10337.

One male in ethanol with a drop of glycerol deposited at El Colegio de la Frontera Sur, Unidad Chetumal, Access number ECO-CH-ZPL10338.

**Diagnosis.** Medium-sized daphniids. Length of parthenogenetic females: 541 ± 13.3 μm ($n = 6$), height 342.6 ± 23.8 μm ($n = 6$), body ovoid. Ventro-posterior corner of valves angular, forming a well-defined mucro of medium length (Figs. 1A–1D). Ephippium slightly reticulated with a strong horse saddle shape and a single egg (Fig. 1B). Large and bilaterally ridged head with a flat surface. The head with a strong reticulation (Figs. 1E and 1F), but valves smooth in larger females. Rostrum trilobate, the middle lobe well developed with sides straight to relatively rounded (Figs. 1G and 1H), the frontal head pore with an elongated shape. Antennule short, with nine terminal aesthetascs and a lateral sensory seta (Fig. 1H). Antenna with a four-segmented exopod and a three-segmented endopod. Posterior rim of valves with a stronger seta in the last third (Fig. 2D), with a single membrane denticulate all its way (Figs. 2D and 2E). The ventral rim of valves infolded, forming an adhesive sucker-plate (Fig. 3A), that in the middle zone present lamellae internally forming triangle-like structures ending in three long filaments (Figs. 3C and 3D) and externally have indented convex folds but with spine-like projections in the free border (Fig. 3F). Five trunk limbs of general daphnid shape. Endopodite of trunk limb I with a brush-like seta with the fine hairs in the top decreasing

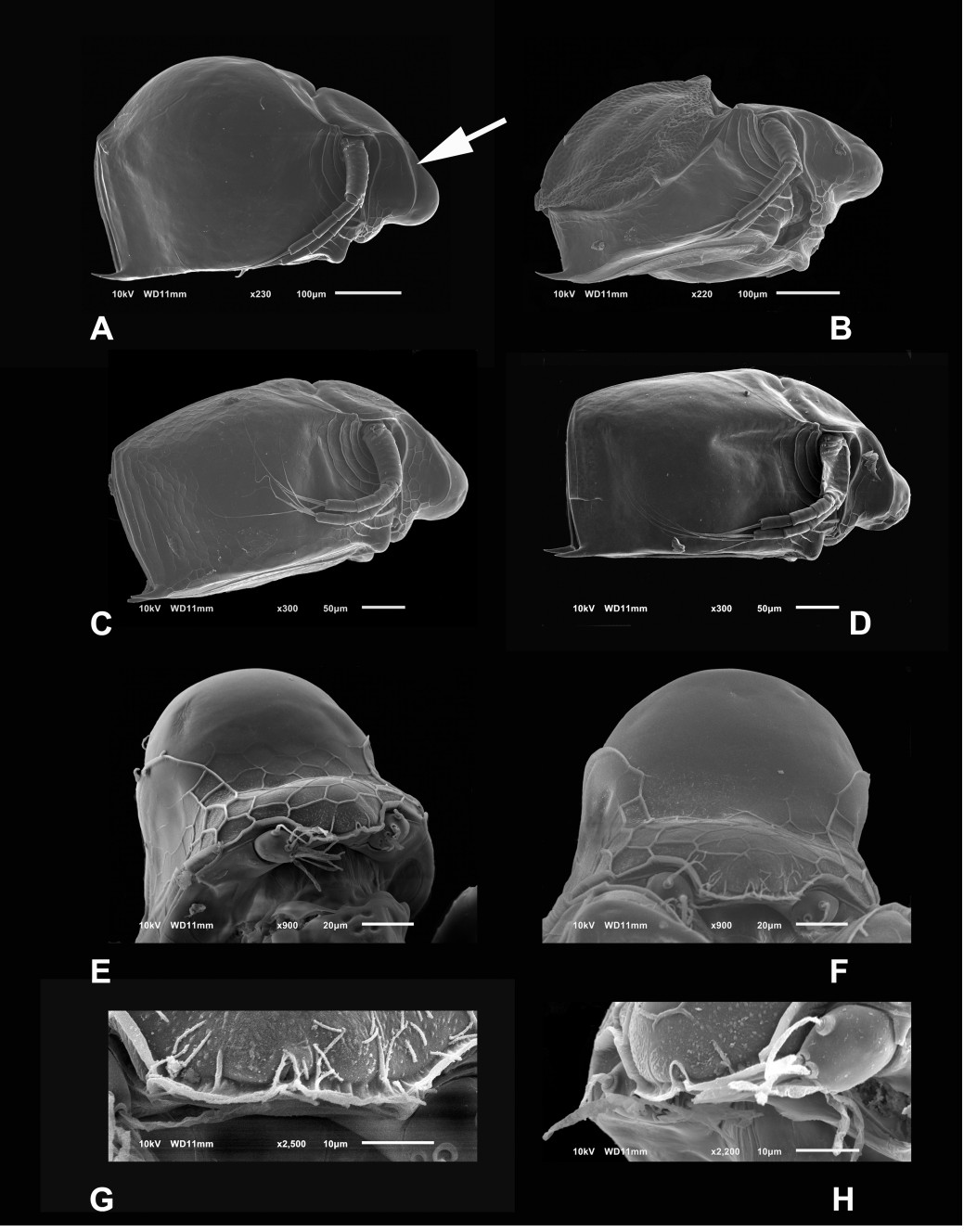

**Figure 1 SEM photographs of *Scapholeberis yahuarcaquensis* n.sp. from near San Antonio village, Quebrada Yahuarcaca.** (A) Adult parthenogenetic female, arrow points sclerotized ridge. (B) Ephippial mature female. (C) Immature female. (D) Immature female. (E) Head, adult. (F) Frontal view showing the frontal head pore. (G) Close view of the frontal head pore in F. (H) Immature female, pore and A1.

in size, giving a finer tip than the base (arrow, Fig. 4F). Postabdomen robust, more or less rectangular, with dorsal margin flat and straight, is covered by rows of spinules (Fig. 5C). Postabdomen claw with three successive pectens of different sizes on each internal and external margin (Figs. 5B–5E). Male with ventral sucker-lamellae much more

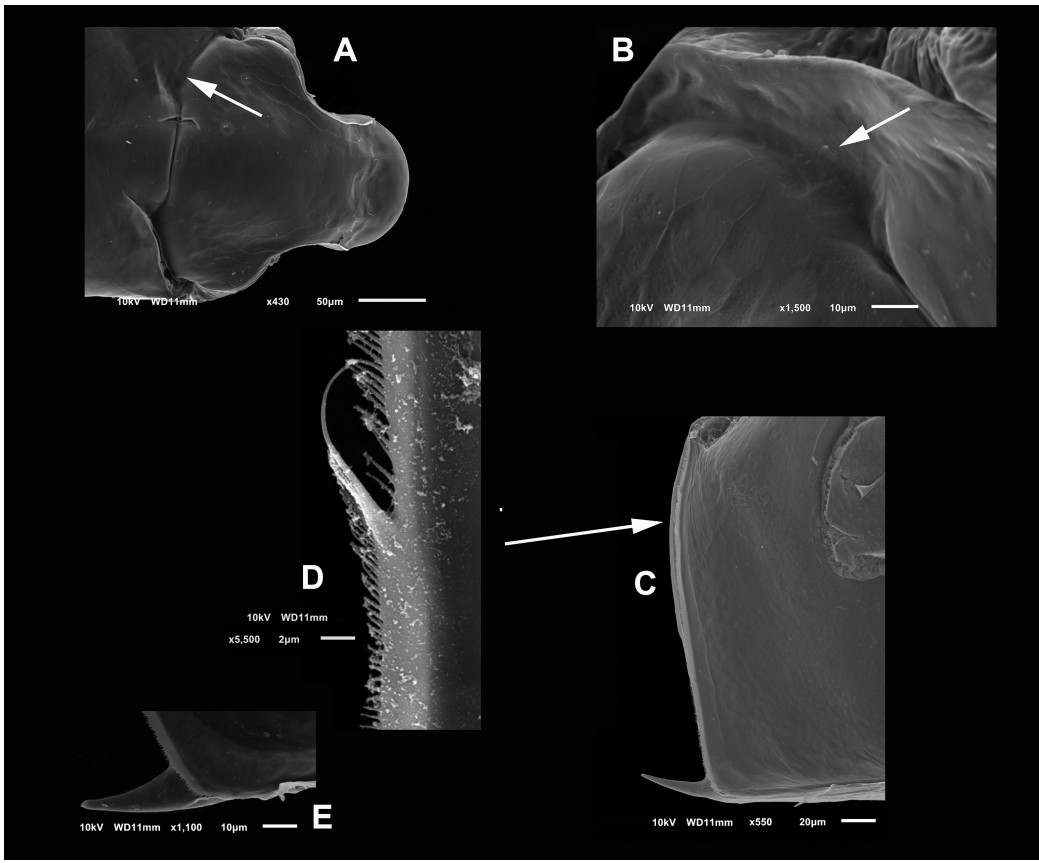

**Figure 2** (A and B) SEM photographs of *Scapholeberis yahuarcaquensis* n.sp. from Tracacha Renaco, Quebrada Yahuarcaca. (C–E) SEM photographs of *Scapholeberis yahuarcaquensis* n.sp. from near San Antonio village, Quebrada Yahuarcaca. (A) Parthenogenetic female, dorsal view showing pore position in one side (arrow). (B) Close view of the pore. (C) Close view of the seta in the hyaline lamella. Arrow points its location in posterior margin. (D) Internal view of the hyaline lamella along the posterior margin of the valve. (E) Close view of the origin of the hyaline lamella, near the mucro.

developed than female (Fig. 6A) and rectangular shape of the body. Male hook in limb I with a blunt tip and a peculiar lamella-like outgrows in the proximal side. Paddle with well-developed spines scattered on its surface (Figs. 6C and 6D).

**Description of parthenogenetic female.** Head similar to *S. duranguensis* (see Figs. 1E and 1F). Middle lobe of rostrum well-developed straight to slightly rounded with an elongated slit-pore structure (the so-called frontal head pore) at the tip (Figs. 1E–1H). Globular eye and round ocellus.

**Antennule** short and oval with nine terminal aesthetascs cylindrical and different length (Figs. 1H and 4A). With a lateral sensory seta of the same size as the aesthetascs.

**Head** smooth in adult females, slightly reticulated in juveniles, delimited by two sclerotized ridges that extend to each side of the head (Fig. 2A, arrow in Fig. 1A). In the headshield's

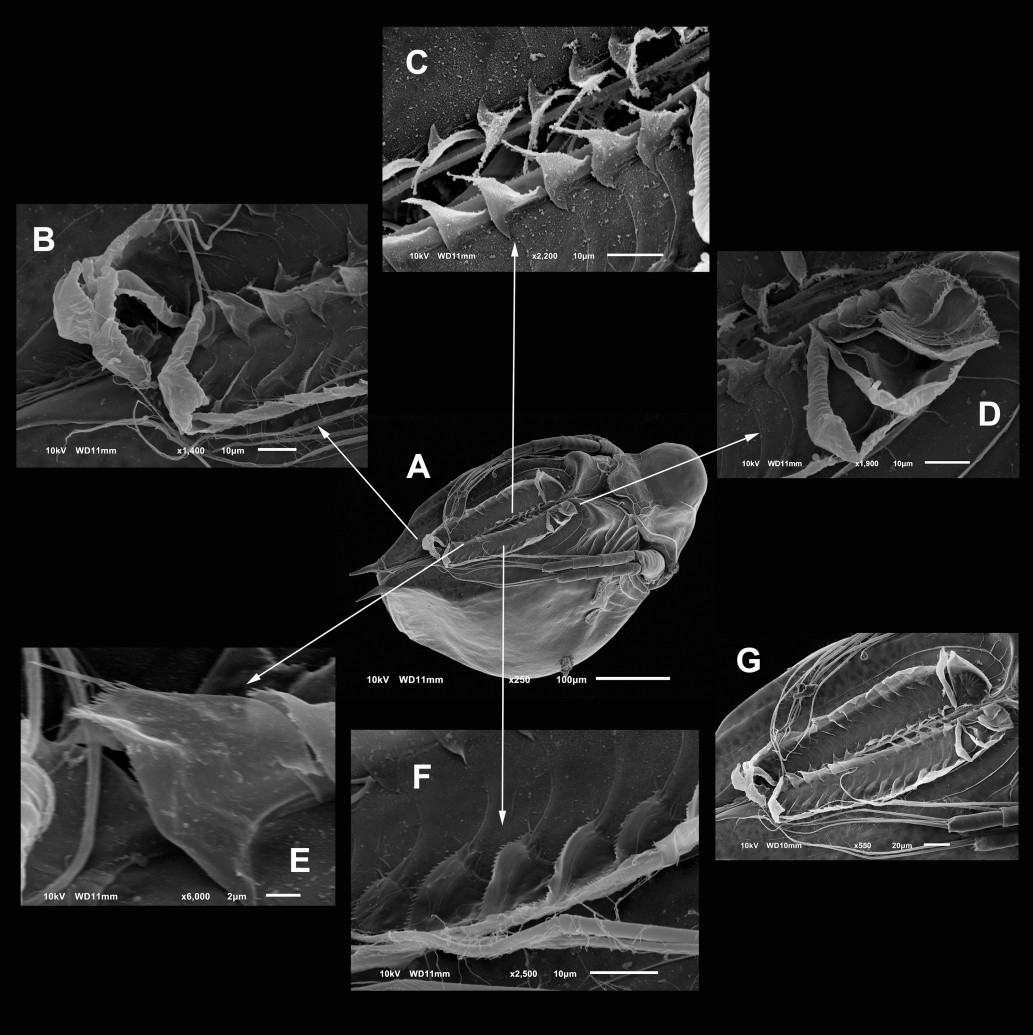

**Figure 3** SEM photographs of the ventral sucker of *Scapholeberis yahuarcaquensis* n.sp. from near San Antonio village, Quebrada Yahuarcaca. Arrows point to the enlarged region of the sucker from the ventral view of a mature parthenogenetic female (A). (B) Lamellae in posterior part. (C) Lamellae in the middle part. (D) Lamellae in the anterior part. (E) Denticled lamellae in the posterior third. (F) Lamellae in the outer side in the middle part of the sucker. (G) Close view of the whole sucker.

junction with the valves, there are two small lateral pores on each side (arrow in Figs. 2A and 2B).

**Valves** in juvenile animals sometimes with reticulation, mostly in the posterior-dorsal regions (Fig. 1C). Other specimens with a smooth surface (Fig. 1D). Adult parthenogenetic females with smooth valves, and some reticulation and parallel lines near the A2 (Fig. 1A). The posterior margin of the valves with a single denticulated lamella (Fig. 2C), forming in the dorsal side a wider seta (Fig. 2D). In the ventral part, near the mucro, the hyaline lamella is formed by a fold of the sucker (Fig. 2E). Ventral rim of valves infolded, supplied with lamellae well developed that form an adhesive sucker-plate (Fig. 3A). The lamellae in the sucker-plate different in three clearly defined regions: On the

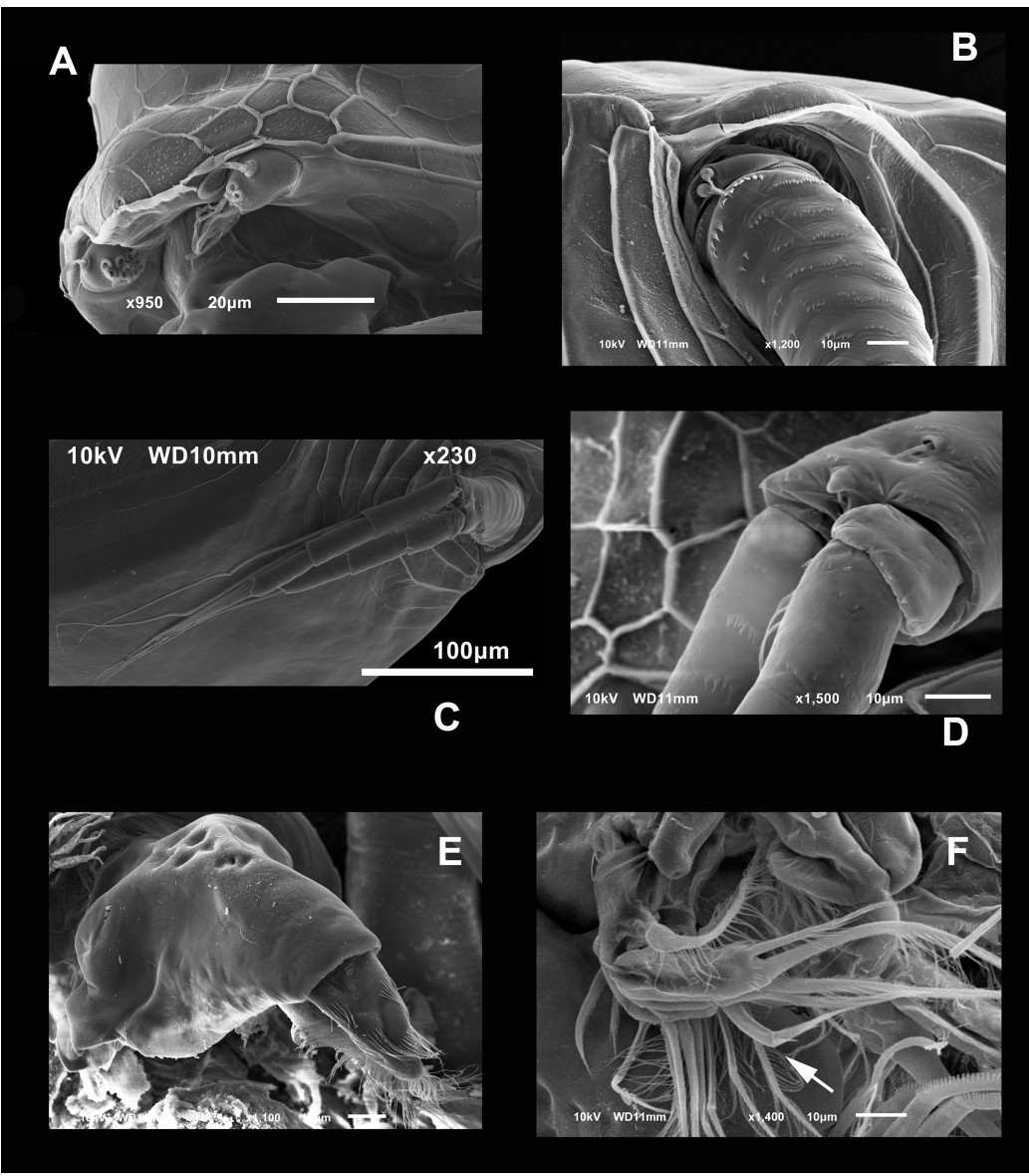

**Figure 4** **SEM photographs of *Scapholeberis yahuarcaquensis* n.sp. from near San Antonio village, Quebrada Yahuarcaca, female.** (A) Close view of the rostrum and A1. (B) Basipodite and two setae near the external base. (C) General view of the A2. (D) Distal side of the basipod, with finger-like projection and the seta emerging from a pore. (E) Labrum. (F) Limb I, arrow points to the brush-shaped seta.

posterior side, the lamellae are thin and again cross from side to side the sucker from each valve (Fig. 3B). In the middle part of the sucker, the lamellae are divided internally, forming triangle-like structures with three long filaments in the apex (Fig. 3C). Anteriorly, the lamellae are denticulated and cross entirely from inside to the border (Fig. 3D). In the last third of the sucker, the apex of the lamellae widens in a straight end, with a series of long denticles and a filamentous projection on the inner side (Fig. 3E). Externally there are convex folds full of spine-like projections in the free border (Fig. 3F).

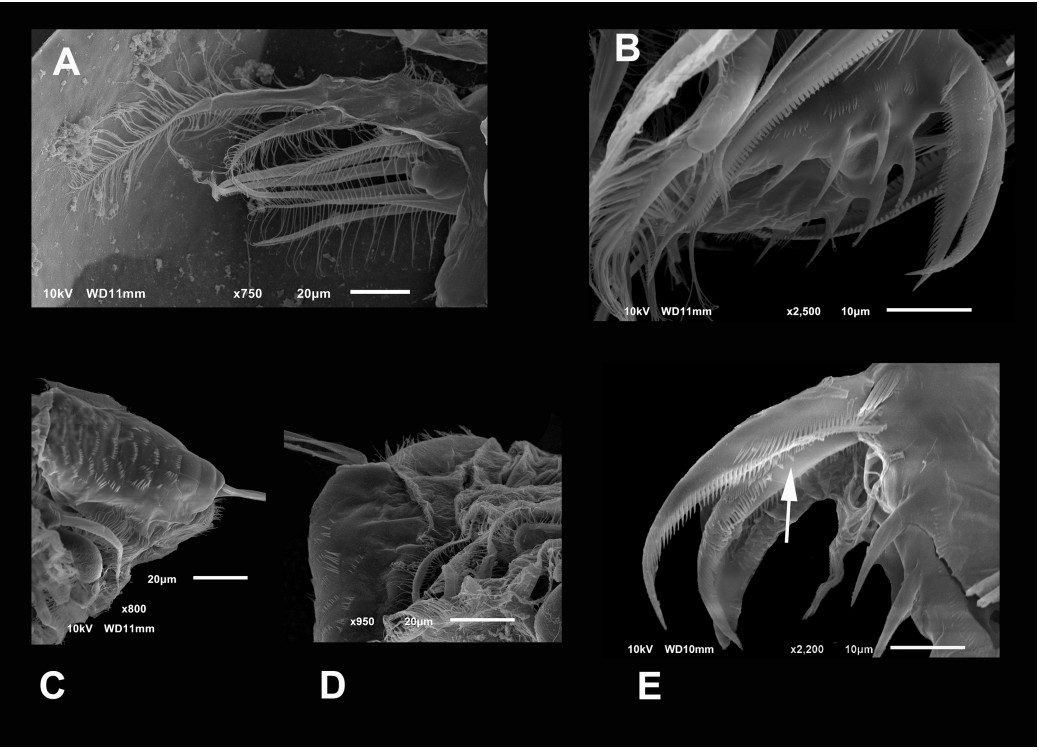

**Figure 5** SEM photographs of *Scapholeberis yahuarcaquensis* n.sp. from near San Antonio village, Quebrada Yahuarcaca, female. (A) P2, exopodite and part of the endopodite. (B) Distal side of post-abdomens and claws. (C) Dorsal view of the postabdomen, postanal region. (D) Dorsal process in the postabdomen. (E) Claws, arrow points to the isolated spines in the inner margin.

**Antenna** with four-segmented exopod and three-segmented endopod (Figs. 1A–1D and 4C). Basipodite thick, with several rows of spines around it, the most proximal bigger, two setae in the outer side of the base (Fig. 4B). In the distal side, there is a pore giving origin a thin seta and a finger-like apical projection (Fig. 4D). In the middle of the exopod's insertion and endopod to the basipod, there is a small projection with a seta. First exopod segment with approximately one-third of the length of the second segment. The latter with three setae in the ventral face and a small terminal seta-like projection. Second segment much shorter than the next Last segment longer than the penultimate (Fig. 4C). First endopod segment much longer than the other two. First and second endopod segments with a long internal-lateral seta. Antennal formula 0-0-1-3/1-1-3 setae, spines 0-1-0-0/0-0-0.

**Labrum** triangle-like shaped, with the anterior margin pitted, with the tip covered by lines of hairs (Fig. 4E).

**Postabdomen** rectangular with ventral margin convex (Figs. 5C and 6E). Length approximately three times more than height. Preanal portion approximately one-third of the total length. Ventral face with two lateral rows of four spines, decreasing in size proximally. Dorsal face straight with several rows of strong spines (Fig. 5C).

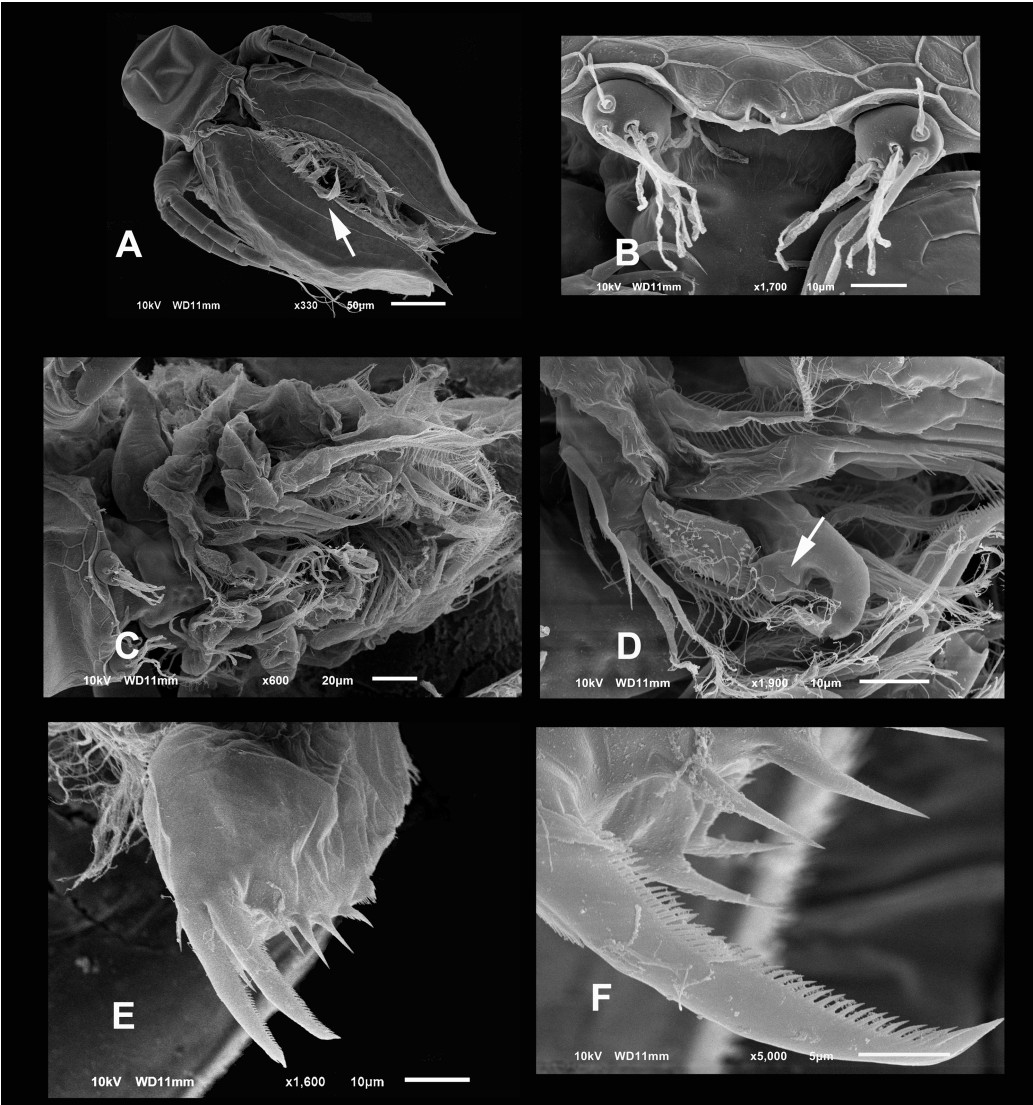

**Figure 6** SEM photographs of *Scapholeberis yahuarcaquensis* n.sp. from Tracacha Renaco, Quebrada Yahuarcaca, Male. (A) Ventral view, arrow points to long lamellae from the anterior part of the sucker. (B) Close view of the rostrum and A1. (C) Arrangement of the limbs, in the left side the well-developed hooks of P1 are visible. (D) Close view of the male hook and the pad (body of endopodite). Arrow points the triangle-like lamellae on the basis of the hook. (E) Ventral view of the postabdomen. (F) Claw.

**Postabdominal claw** shorter than the preanal portion (Fig. 5E). With six rows of spinules (pectens) on each claw. On the most outer side, near the base of the claw, there are 4–5 thick and prominent spines forming a group decreasing in size proximally (arrow in Fig. 5E). The basal internal pecten, ascends in a dorsal direction, up to a fifth of the beginning of the medium pecten, which is similar in size and shape of the spines. The medium external pecten starts at the base of the claw and extends to a bit less than half of it. The internal one is similar, but the spines are a little shorter and thinner. The longer external pecten has spines gradually increasing slightly in size, starts at the last third of

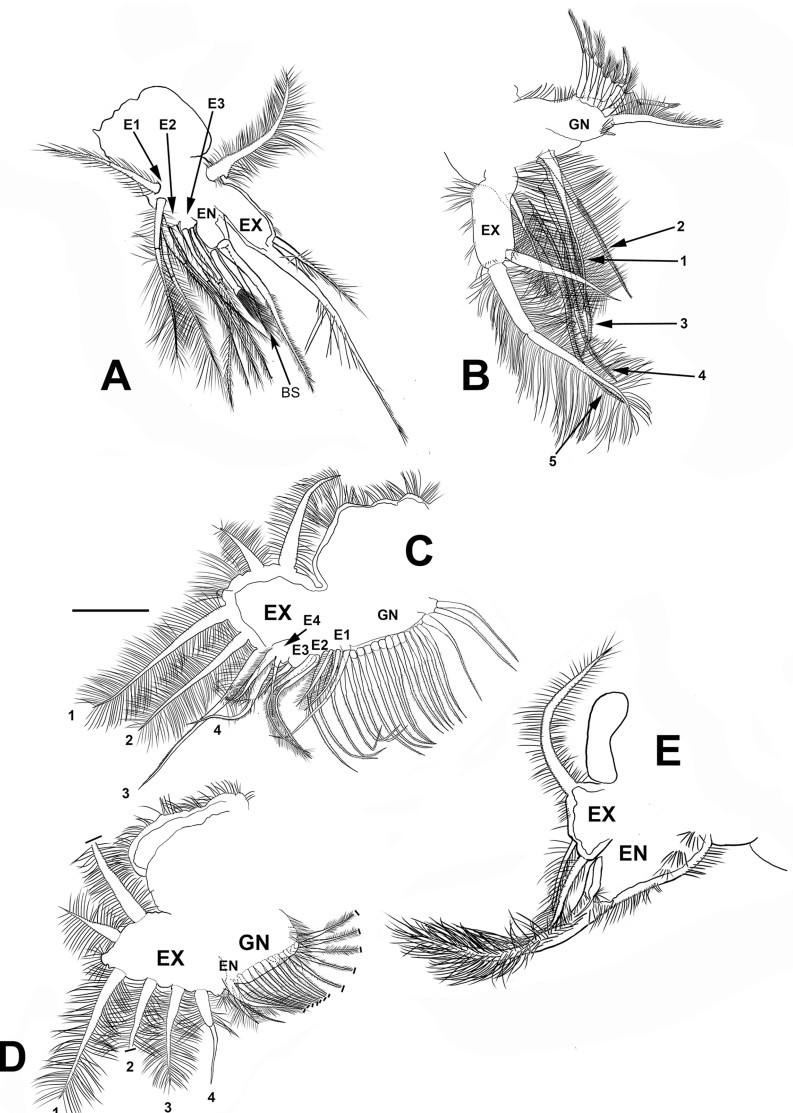

**Figure 7** *Scapholeberis yahuarcaquenis* n. sp. from Quebrada Yahuarcaca, near village San Antonio, **female.** (A) Trunk limb I. (B) Trunk limb II. (C) Trunk limb III. (D) Trunk limb IV. (E) Trunk limb V. Initials: EN, endopod; EX, exopod; E, endite; GN, gnathobase; BS, brush-shaped setae. The scale bar is 67 μm for (A) and (E); 50 μm for all others.

the mid pecten, and extends to the claw's tip. The internal one starts with 6–7 single wide spines (arrow) and continues with the spines gradually decreasing in width to the tip.

**Trunk limb I** (P1). Exopod with two unsegmented, apical setae, unequal in size (Fig. 7A). With one row of long and thick setulae in the distal two-thirds of both. In the middle of the longer seta, a bunch of short and thick setulae. A long accessory seta at the basal external face of the exopod, with a triangular projection near its base. Endite 4 with three apical setae. The longest seta with short setulae bilaterally arranged from the half to the tip. The second seta about two-thirds as long as the first one and with an apical tuft of long setulae. The third seta is brush-shaped. This seta has at the tip setulae, increasing

their length towards one end, giving a peculiar form (Figs. 7A and 4F, arrow). Endite 1, and endites 2 and 3 similar to *S. duranguensis*. The most proximal seta shorter and bilaterally setulated. Ejector hooks long, with setulae along their inner side (Fig. 4F).

**Trunk limb II** (P2). Exopod, endopod and gnathobase similar to *S. duranguensis* (Fig. 7B). The two appendages found in the base of the exopod in *S. duranguensis* seem to be absent.

**Trunk limb III** (P3). Exopod also similar to *S. duranguensis* (Fig. 7C), except for the following: setae 3 and 4 with fine spines in one face, and the last two thirds with short, strong setae. Seta 4 with a crown of long and thick setae. Endite 1 with long and naked seta and two shorter and bilaterally setulated setae. Endite 2 with two long setulated setae. Endite 3 with one strongly naked chitinized seta and one normal setulated seta. Endite 4 with a similar arrangement as E3, but both elements longer. Gnathobase with 18, bisegmented and thin setae similar in size and shape. At the internal end, there is a lobe with two additional setae.

**Trunk limb IV** (P4). This limb is similar to other *Scapholeberis* (Fig. 7D), in particular, it resembles the detailed description of *S. duranguensis* by *Quiroz-Vazquez & Elías-Gutiérrez (2009)*.

**Trunk limb V** (P5). With a long, bisegmented, densely ciliated seta in the endopod (Fig. 7E), another thinner and shorter behind it. A tiny structure at the base of the seta with a bunch of setulae and distal tip narrower is approximately one-third of the total long seta. Exopod with two shorter setulated setae. There is a long setulated distal seta.

**Description of ephippial female.** Carapace less massive than egg-carrying females (length 527.7 ± 31.6 μm, *n* = 3), height 310.5 ± 39.5 (Fig. 1B). Ephippia thick, convex in the dorsal keel, with a strong projection towards the front and the back. The surface of the ephippia pitted and covering about 2/3 of the whole carapace. The ephippial females have the carapace brown and dark brown towards the dorsum. All other characters are similar to the parthenogenetic females.

**Description of male.** With strong sexual dimorphism, males are smaller with length 371 ± 11.7 μm (*n* = 3), height 188 μm (*n* = 2), rectangular in shape, similar to immature females (Figs. 1C and 1D), but the lamellae of the sucker much more developed than in females (Fig. 6A, arrow). A1 with longer asthetascs (Fig. 6B) and P1 strongly modified with a hook in limb I. The hook has a blunt tip, and a triangular lamella-like outgrows in the proximal side (Fig. 6D). Copulatory pad with strong spines arranged in lines, two parallel in distal part (Fig. 6D).

**Postabdomen** less robust than females, with fewer spines and shorter. With no genital pores apparent (Fig. 6E).

**Postabdominal claw** similar to females, but the most internal pecten lacks the single robust spines. The group spines on the outer side of the basis of the claw have only three spines more strongly fuzed than in females (Figs. 6E and 6F).

All other characters are similar to females.

**DNA Sequences (CO1):**

We got nine almost identical sequences, representing two possible haplotypes, with an overall distance of 0.3% from parthenogenetic and ephippial females from two close points in the Amazon basin (see type material). All of them are in the dataset DS-SCAMACOL in BOLD. A consensus sequence for this species is:

| | | | | | |
|---|---|---|---|---|---|
| CATTATATTT | TATTTTTTGG | TGTTTGATCT | GGTATAGTTG | GGACCGCTCT | 50 |
| TAGTATGCTA | ATTCGGGCAG | AGTTGGGCCA | GGCTGGTACT | TTAATTGGGG | 100 |
| ATGATCAGAT | TTATAACGTA | GTTGTTACAG | CCCATGCTTT | TGTAATAATT | 150 |
| TTCTTTATGG | TTATACCAAT | TATAATTGGA | GGTTTTGGGA | ACTGGCTAGT | 200 |
| TCCTTTAATG | TTAGGTGCTC | CCGATATAGC | TTTTCCTCGA | TTAAATAACC | 250 |
| TAAGTTTTTG | ATTTCTTCCT | CCTGCCTTAA | CTCTTCTTTT | AGTTGGAGGG | 300 |
| GCGGTAGAAA | GTGGGGCTGG | AACTGGCTGA | ACTGTTTATC | CTCCTTTATC | 350 |
| GGCGGGAATC | GCTCACGCTG | GAGCCTCAGT | CGATTTAAGA | ATCTTCTCCC | 400 |
| TTCATTTGGC | AGGGATTTCT | TCTATTTTAG | GAGCTGTTAA | TTTCATTACT | 450 |
| ACTATTATTA | ATATACGATC | GGAAGGAATG | TCTCTAGATC | GAATTCCATT | 500 |
| ATTTGTGTGG | GCGGTAGGAA | TCACAGCTCT | CCTTCTACTT | TTAAGTCTTC | 550 |
| CTGTTTTAGC | AGGGGCAATT | ACGATGCTTC | TAACGGATCG | GAACTTAAAC | 600 |
| ACCTCGTTTT | TTGACCCGGC | TGGAGGAGGA | GATCCAATTC | TCTATCAACA | 650 |

GC% of the first and third codon positions were similar, with a percentage ranging from 53.00 to 53.46 and 25.46 to 25.93. A% and T% gave the main total variation.

DNA analyses revealed 11 clades with the COI gene (Fig. 8). Clade for *S. yahuarcaquensis* n.sp. is well supported by the Bayesian analyses of coalescence (Supplemental File 2) and the Barcode Index Number (BIN) (ADN0437) (*Ratnasingham & Hebert, 2013*). The closest species to *S. yahuarcaquensis* n. sp. is *S. freyi* from Mexico with a mean distance 16.4% divergence, followed by *Scapholeberis* sp. (Chihuahua) and *S. duranguensis Quiroz-Vazquez & Elías-Gutiérrez, 2009*, both from Mexico with 18.4% and 18.7% divergence (Table 1). A big not solved group is the northern America named *Scapholeberis armata* group, represented by four BINS distributed from the north of the United States to Canada and Alaska. All other *Scapholeberis* from the world form definite clades, but the species name in some cases as *Scapholeberis rammneri* s.l. require further studies.

## DISCUSSION

Records of *Scapholeberis* species in America are mainly from Mexico, the United States, and Canada. In South America, there were only five species and one subspecies registered: (1) *Scapholeberis armata* Herrick, 1882, is a Nearctic species reported in Colombia for the Andean region (*Kotov & Fuentes-Reines, 2015*). (2) *Scapholeberis mucronata* (O. F. Müller, 1776) registered for the Nearctic, Palearctic, Neotropical regions (*Kotov et al., 2013*) and Colombia is cited in the Caribbean (*Kotov & Fuentes-Reines, 2015*). (3) *Scapholeberis kingi* Sars, 1888, a Palearctic species, according to *De los Rios-Escalante & Kotov (2015)*. The neotropical records probably correspond to *S*. cf. *rammneri Dumont & Pensaert (1983)* (*Taylor, Connelly & Kotov, 2020*). (4) *S. rammneri*, which, according to

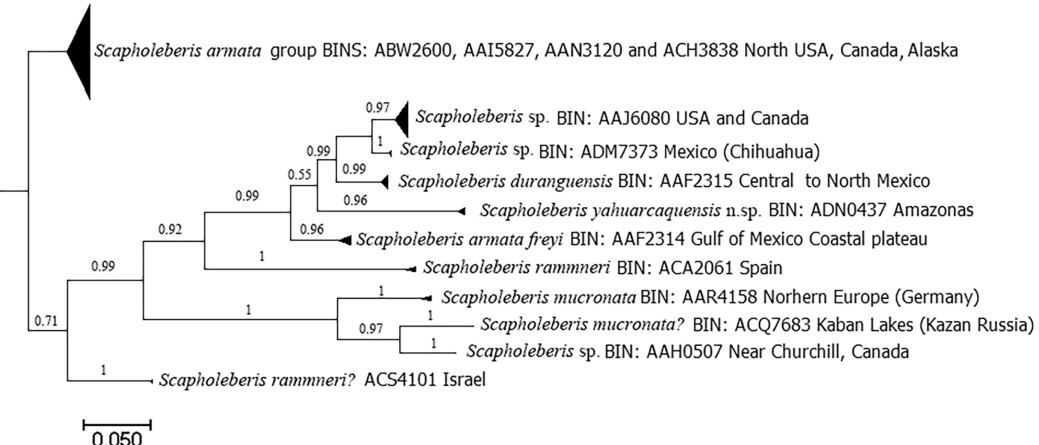

**Figure 8 Bayesian tree after the molecular analysis of the COI gene comparing the known sequences for *Scapholeberis* in BOLD.** Names and interim names are as they appear in the databases. At the end of each clade is shown the BIN number given by BOLD system and the place of collection.

**Table 1 Shows mean divergences among groups of *Scapholeberis*.**

|   | A | B | C | D | E | F | G | H | I | J | K |
|---|---|---|---|---|---|---|---|---|---|---|---|
| A |   | 0.021 | 0.023 | 0.020 | 0.022 | 0.022 | 0.022 | 0.018 | 0.023 | 0.022 | 0.025 |
| B | 0.212 |   | 0.025 | 0.022 | 0.023 | 0.024 | 0.015 | 0.015 | 0.010 | 0.020 | 0.023 |
| C | 0.244 | 0.275 |   | 0.025 | 0.022 | 0.019 | 0.027 | 0.023 | 0.027 | 0.025 | 0.025 |
| D | 0.182 | 0.225 | 0.269 |   | 0.025 | 0.025 | 0.025 | 0.023 | 0.024 | 0.025 | 0.026 |
| E | 0.244 | 0.248 | 0.185 | 0.248 |   | 0.017 | 0.024 | 0.023 | 0.024 | 0.024 | 0.026 |
| F | 0.248 | 0.262 | 0.161 | 0.259 | 0.132 |   | 0.024 | 0.023 | 0.023 | 0.024 | 0.027 |
| G | 0.218 | 0.141 | 0.283 | 0.249 | 0.242 | 0.257 |   | 0.015 | 0.015 | 0.019 | 0.023 |
| H | 0.185 | 0.136 | 0.263 | 0.233 | 0.248 | 0.260 | 0.134 |   | 0.015 | 0.019 | 0.021 |
| I | 0.231 | 0.068 | 0.289 | 0.229 | 0.249 | 0.246 | 0.121 | 0.128 |   | 0.022 | 0.024 |
| J | 0.223 | 0.188 | 0.265 | 0.262 | 0.263 | 0.238 | 0.164 | 0.187 | 0.184 |   | 0.021 |
| K | 0.268 | 0.233 | 0.271 | 0.268 | 0.274 | 0.289 | 0.208 | 0.220 | 0.229 | 0.208 |   |

**Note:**
Estimates of evolutionary divergence over sequence pairs between groups . Standard error estimates are shown above the diagonal. (A) *S. armata* group North America; (B) *Scapholeberis* sp. USA and Canada; (C) *S. mucronata* Northern Europe; (D) *S. rammneri* Israel; (E) *S. mucronata* Kaban Lakes; (F) *Scapholeberis* sp. Churchill; (G) *S. freyi* Gulf of Mexico; (H) *S. duranguensis*; (I) *Scapholeberis* sp. Chihuahua; Mexico; (J) *S. yahuarcaquensis* n. sp.; (K) *S. rammneri* Spain.

*Taylor, Connelly & Kotov (2020)*, is widespread in South America. (5) *Scapholeberis spinifera* (Nicolet, 1849) of the most southern portion of South America (*Taylor, Connelly & Kotov, 2020*). (6) *Scapholeberis armata freyi Dumont & Pensaert, 1983*, listed as *Scapholeberis freyi Dumont & Pensaert, 1983* by *Kotov et al. (2013)*, and with Nearctic and Neotropical distribution.

A problem is to collect these hyponeuston taxa with conventional methods as plankton tows that usually pass below the surface of the water. They are considered a "bycatch" from other methods targeting immature mosquitoes (*Taylor, Connelly & Kotov, 2020*).

We found that *Scapholeberis* has a strong positive phototaxis on all stages of the life cycle, particularly males and ephippial females are scarce in "normal" sampling, possibly due to their rarity in the environment. After only 3 h of work with the light traps, we obtained an ample sample with positive records in three localities. These samples had not only *Scapholeberis* representatives. It was a real "soup" of zooplankton. We believe that using light traps will help understand the diversity of zooplankton in this important region.

Of the known taxa for South America, only in two cases material collected in the Neotropical zone has been included for the descriptions, *S. spinifera*, and *S. freyi*, the latter from the material of Paraguay from Daday's collection (*Dumont & Pensaert, 1983*). In all the other cases, the reports in South America have been assumed morphologically consistent with the species described for other latitudes or other continents.

We consider that detailed morphological analysis and the use of molecular tools, like DNA barcoding, will help to uncover the real identity of all these species, as the case presented here or with *Scapholeberis duranguensis Quiroz-Vazquez & Elías-Gutiérrez, 2009*, which was initially cited as *S. rammneri* by *DeWaard et al. (2006)* and for which the closest species was *S. freyi*. The GMYC model used as a species delimitation method has been tested in other crustaceans as copepods (*Gutierrez-Aguirre et al., 2020*). It has been consequent with other methods as the BIN system and morphology, giving strong support to the modern description of species.

*Scapholeberis freyi* is distributed from Florida to the south coast of the Gulf of Mexico in the neotropics. In South America, this species has been reported in Brazil (*Elmoor-Loureiro, 2000*) and Paraguay (*Dumont & Pensaert, 1983* and references therein). *De Abreu (2016)* compared a sequence (KU315490) from Minas Gerais (Brazil) with all available sequences in BOLD and GenBank, and she found sharp divergences with all of them. She concluded that possibly her specimen belongs to a different species, but the name found in GenBank still is *S. armata freyi* (see https://www.ncbi.nlm.nih.gov/nuccore/KU315490.1). Our sequences were grouped with this one (Fig. 8, Supplemental File 2), so we can conclude that these specimens were conspecific and the distribution of the *S. yahuarcaquensis* n.sp. extends beyond the Amazon basin. In total, three haplotypes represented this species (Supplemental File 2). This sequence had only 92% similarity with the closest match of *Taylor, Connelly & Kotov (2020)* comparisons with COI.

A comparison with *S. freyi*, the closest relative of *S. yahuarcaquensis* n.sp. reveals important differences in morphology: rostrum less wide and almost straight in adult females with an oval pore (similar to *S. kingi*, see Plate 3–7 in *Dumont & Pensaert, 1983*). The dorsal pores are at the juncture of the cephalic shield and valves as in *S. kingi* from Australia (*Dumont & Pensaert, 1983*). The number of spines in the base of the claw at the outer side is more numerous.

The sucker is quite peculiar in *S. yahuarcaquensis* n. sp., with the lamellae forming delicate triangles with filament-like projections (Fig. 3C). Another feature unique in the sucker region is the external lamellae with denticle-like projections. The male has a strong development of the sucker-lamellae in the anterior end and diminishing in size

posteriorly (see Fig. 6A) opposite as described for the male of *S. freyi* by *Dumont & Pensaert (1983)*. All setae and aesthetascs have a pore in the A1. The hook in male P1 is not sharply recurved as *S. freyi*, and the pad has a unique armature with the spine-like projections as described previously.

Although some similarities are found between *S. kingi* from the old continent and *S. yahuarcaquensis* from South America, differences are given by the middle lobe of the rostrum almost straight, the ventral sucker, valves with no sculptures in adult females, with a single hyaline membrane with different arrangement and not broad. The male has a middle lobe in the rostrum wide and flat in *S. yahuarcaquensis*, not short. The hook in P1 has four lamella-like projections triangle-shaped, not two strong teeth, and several rows of spinules in the pad and the ventral sucker is entirely different, much more developed.

## CONCLUSIONS

This study is an example of a description based on all stages of the life cycle of a cladoceran. In most taxonomical analyses, the collection of males or ephippial females is difficult because they are mostly rare. This case is the first of new and exciting discoveries in the Amazon basin by using complementary methods as plankton tows and light traps that never before were used here.

Finally, long ago, *Dumont & Pensaert (1983)* pointed out that several undescribed species of *Scapholeberis* await their discovery in South America. We consider that more extensive samplings are urgent in this crucial region to continue the documentation of their biodiversity due that most of their aquatic environments are under a severe threat.

Once we have proper documentation in public databases of the species present, we could use new technologies currently available as next-generation sequencing and environmental DNA. Zooplankton community is the first to shift due to any change that could be subtle as the translocation of species or significant as the enrichment of the waters after extensive use of fertilizers, as nowadays occurring in several parts of the Amazon basin.

## ACKNOWLEDGEMENTS

Professor Santiago R. Duque (Instituto Amazónico de Investigaciones [IMANI], Universidad Nacional de Colombia—Amazonian Campus) for his support of field activities in the Amazonian region. Alma Estrella García Morales from Chetumal Node of the Mexican Barcode of Life Network (MEXBOL) performed all DNA analyses. Osvar Cupitra-Gómez, Alberto Moncayo-Fernández, María del Mar Rivera-Portilla and Juan Monteiro kindly assisted us with the field work. The people from the village La Playa supported our work and allowed us to sample in their region. This work is part of the results of the doctoral thesis of CA-S and it was developed within the framework of an international agreement signed between El Colegio de la Frontera Sur and Universidad del Cauca. ME-G recognizes the high authorities of the Mexican government for their effort to smother all aspects of science progressively, particularly in regard to international cooperation and biodiversity conservation, as a motivation to produce and conclude this work.

### Funding
This study was financed by Universidad del Cauca and Universidad Nacional de Colombia. The molecular part of this work was supported by MEXBOL network through El Colegio de la Frontera Sur. The funders had no role in study design, data collection and analysis, decision to publish, or preparation of the manuscript.

### Grant Disclosures
The following grant information was disclosed by the authors:
Universidad del Cauca and Universidad Nacional de Colombia.
MEXBOL network.

### Competing Interests
The authors declare that they have no competing interests.

### Author Contributions
- Camilo Andrade-Sossa conceived and designed the experiments, analyzed the data, prepared figures and/or tables, authored or reviewed drafts of the paper, and approved the final draft.
- Lorena Buitron-Caicedo conceived and designed the experiments, analyzed the data, prepared figures and/or tables, authored or reviewed drafts of the paper, and approved the final draft.
- Manuel Elías-Gutiérrez conceived and designed the experiments, performed the experiments, analyzed the data, prepared figures and/or tables, authored or reviewed drafts of the paper, and approved the final draft.

### Field Study Permissions
The following information was supplied relating to field study approvals (i.e., approving body and any reference numbers):
Field permit was issued by Autoridad Nacional de Licencias Ambientales (ANLA) (Resolución #0152).

### DNA Deposition
The following information was supplied regarding the deposition of DNA sequences:
Barcode of Life Database: DS-SCAMACOL
DOI 10.5883/DS-SCAMACOL
GenBank: MT607962–MT607970

### Data Availability
Sequences and species suggested by coalescence are available in the Supplemental Files.

## New Species Registration

The following information was supplied regarding the registration of a newly described species:

Publication LSID: urn:lsid:zoobank.org:pub:5CF5417E-A35F-42BC-97D4-5BB4AD6D6259.

Scapholeberis Schoedler, 1858 LSID: urn:lsid:zoobank.org:act:AB6A5869-8F86-47EA-B1F1-A36ECADB3AE8.

## Supplemental Information

Supplemental information for this article can be found online at http://dx.doi.org/10.7717/peerj.9989#supplemental-information.

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
