# Peer review of "A new species of Scapholeberis Schoedler, 1858 (Anomopoda: Daphniidae: Scapholeberinae) from the Colombian Amazon basin highlighted by DNA barcodes and morphology"

_PeerJ, doi:10.7717/peerj.9989_

## Round 0.1 · original submission · Minor Revisions

Thank you very much for the nice research. I and three reviewers have now completed the review of this manuscript. You had provided the discussion of the Scapholeberis species of Colombian Amazon Basin, and describe a new, Scapholeberis species. I think this manuscript had high potential to published in PeerJ if you revised the manuscript following three reviewers. I am waiting for your revised manuscript.

·

Basic reporting

Paper is in general OK and could be published after some corrections. See comments to the authors.

Experimental design

The study did not contain the experiments.

Validity of the findings

The conclusion is properly confirmed.

Additional comments

I can recommend to publish this paper. At least the last author has a necessary qualification for an adequate species description. The paper agrees with the taxonomic rules (but see comments below), the taxon name is deposited to the Zoobank, and sequences are also properly deposited, and their deposition is clearly noted in the manuscript. Field studies made with all necessary permits.

I can not verify that English is perfect (or not). I recommend to show the paper to native speakers.

Critical comments:

I have no doubts that the authors have really a new species. I did not see their material, but I recommend to diminish the discussion of characters of the ventral portion of carapace. This opinion is based on study of few specimens under SEM, as observation of this portion using optical miscroscope is problematic, as this structures not so transparent. I saw a lot of specimens of Scapholeberis and I argee with the opinion of Dumont & Pensaert (1983) that this portion id very variable among specimens. Moreover, preservation of the membranes in this portion is different in different specimens from same sample due to different level of fixation and destruction of the membranes during the animal life. I saw specimens from same populations with well-preserved and very shabby membranes. Therefore I believe that the differences exist, but I stronglu recomment to the authors to be concentrated on obvious characters on the ventral valve portion, and reduce as much as possible the discussion of the differences in fine characters, because this comparison is based on few animals from few populations.

This is a relative short paper, but its reference list is specially short, what is uncharacteristic for taxonomic papers. In reality, some contribution to the studies of Scapholeberis is made recently by Alonso (1996), Hudec (2010). Moreover, Taylor et al. (2020) concluded that in general the conclusions of morphologist for the taxon delimitation in Scapholeberis were quite correct. I think that the solely morphological systematics must be noted (1 paragraph with some references).

See also a list of particular comments:

9, 37, 57. The first mention of a taxon of generic and species rank must be accompanied by the taxon author(s) name(s) - in the title, the Abstract and the Manuscript body. Also first reference on the genus in the title, Abstract and the main body must be accompanied by explanation of its taxonomic position, preferably with family information ? (Cladocera: Anomopoda: Daphniidae).

53. Abstract is solely descriptive. I recommend to add a final conclusion to the end. At the same time, the differential diagnosis in the Abstract could be shortened.

57 and further. Please check everywhere, are taxon names italized?

70. This "Scapholeberis" is extremely dubious. In reality, no evidences that this is a cladoceran were provided by the authors! Moreover, this "Scapholeberis" has segmented postabdomen impossible for cladocerans. See Van Damme K., Kotov A.A., 2016. The fossil record of the Cladocera (Crustacea: Branchiopoda): Evidence and hypotheses. Earth-Science Reviews 163: 162–189.

134, 136. Please fill the numbers XXXX-XXX!

162-167. Please provide the AUTHORS of each taxon.

169 and further. Incorrectly organized. The type locality is a single water body (not several!), from where the holotype is selected.

171 and further. The symbols of degree is necessary for coordinates.

192-193. The measurements like 544 plus-minus 13.3 have no sense, as an instar variability is strong. Only range will be more realistic.

195 and further "so-called rostral pore". This is apparently an outdated term. This is a frontal head pore, with very clear homology, which could be found in many anomopods, including other daphniids except Daphnia.

198 and further. See general comments on the variability of valve ventral portion armature below.

217. Head shiled. Why not head? Really the daphiiids have no headshield in terms of Fryer (1963). They have head capsule.

216. "Endopod" is not proper term here. Endopod is an internal branch of the biramous limb. Your "endopod" is a single "endite 4 only". In reality, endopod consists of "endites 2-4" in your understanding.

218. "A long accessory seta" is a part of exopod also.

309, 313. See on size range above.

326. Why Molecular sequences (it seems that all genetic sequences are molecular ones)? May be "COI geetic barcoding" is better title?

365. Really not Neotropical one, but endemic of the southern most portion of S America

376. Scapholeberis freyi, not armata freyi.

380. Repetition of previous phrase.

Illustrations: For some reason, the illustrations have a huge black margin. All this "empty" black space needs to be cropped.

Reviewer 2 ·

Basic reporting

Line 71 The statement “it is not possible to have only eight species known around the world” because of a Cretaceous fossil appears overstated. The existing fossil has not been assigned to the Genus Scapholeberis. Also, why can’t widespread ancient groups that have been exposed to extinctions be species-poor? Replacing “it is not possible” with “unlikely” would be more prudent.


There are several sentences where the grammar, typographic errors, or language style should be improved for readability. I have listed a few examples below:


Line 34 rich freshwater should be species-rich freshwater

Line 38 light traps of recent design: recently designed light traps

Line 47 “a filament-like projection” doesn’t match the “lament-like projection” in the first abstract of the .pdf (lament is a typo?)

Line 58 hiponeuston should be hyponeuston

Line 60 pellicle should be surface

Line 61 “The structure was detailed described” omit “detailed”


Line 80 sentence needs rewording “not any barcoding study”

Line 92 realized should be carried out

Line 103 “Normanski in a” should be Normanski microscopy (BX51…)

Line 105 in should be with

Line 110 glass fiber method should be glass fiber filtration method

Line 132 keep to past tense in the methods

Line 142 the best fit should be best fit substitution model

Line 160 space needed after and in “andCLOCKSS”

171-172 add degrees symbols to coordinates


Line 231 in should be on

Line 264 not-segmented should be unsegmented

Line 311 epphipial should be ephippial

Experimental design

Lines 369-371

I like the addition of light traps to sampling, but in the text, the relationship between light traps, phototaxis, and the sexual stages is unclear. Are you suggesting that the sexual stages are more positively phototactic than clonal females? Won’t the sexual stages be absent from most light traps because most species are clonal for much of the year? Please clarify.

Validity of the findings

Line 192 “medium-sized daphniids”. I have never plotted daphniid size distributions. However, 0.541 mm seems small for daphniids as nearly all species (mostly Daphnia and Simocephalus) have upper sizes greater than 1mm and many are greater than 3 mm. Consider using “small-sized”.

Lines 347-349 The authors state that the clade for S. yahuarcaquensis is “well supported” by Bayesian analysis. Please give more details about the quantification of this support. Is there a posterior probability? I can’t find the figure caption for Supplementary File 2. Also, please add support values (posterior probabilities?) to Figure 8. Can you provide more details on how the BIN analysis indicates support for new species?

Lines 351-354

The authors use the “armata group” for a large clade and suggest that it needs further study (see also the label on Figure 8.). However, the sampling locations more closely match the geographic range of S. rammneri than S. armata (according to Pensaert & Dumont 1983 and Taylor et al. 2020). Why is “armata” the most appropriate name for this major group? The database names are a mix of no species names, armata group and D. rammneri.

Additional comments

The manuscript provides a detailed species description of a neglected genus of waterfleas, Scapholeberis, from an understudied geographic region for freshwater biology. The morphological description is very detailed with high quality SEM’s, line drawings of limbs and rare information from the sexual stages. The evidence of this new species is bolstered with DNA sequence information from the DNA barcoding region of mitochondrial DNA. I find the evidence of a new species of Scapholeberis to be compelling and in accordance with new species policies and standards. My suggestions above might improve the manuscript and are minor.

Reviewer 3 ·

Basic reporting

As mentioned in the previous section, give more context on current lit in the introduction.

Experimental design

1. For research ethics purposes, cite (in acknowledgment), where applicable, permits acquired to conduct the field sampling.

2. Please add one table about pairwise COI sequence divergence within the Scapholeberis species analysed.

3. Please add key to the species of Scapholeberis.

4. please add bootstrap in Bayesian tree.

Validity of the findings

It is a nice manuscript. I just suggest to overlying the map in figure I think, it will be easy for the readers to understand.

Additional comments

I enjoyed reading your MS but struggled to easily grasp the details largely because of how results were presented and having had to go back and forth from one figure to another. I think your figures are good but contain too much information and need to be explained for e.g. see sections of the discussion highlighted in the PDF. With a little tweaking of how results are conveyed- I think the paper will be widely read and comprehended easily.

Annotated reviews are not available for download in order to protect the identity of reviewers who chose to remain anonymous.

---

## Round 0.2 · accepted · Accept

Thank you very much for responded all comments, the reviewers and I had clear all about your manuscript. Good luck for the next study.

·

Basic reporting

English is improved. Reference list is adequate. Figure, tableas are OK. The paper is improved

Experimental design

The paper is OK for PeerJ. Ethics is OK.

Validity of the findings

Conclusions are supported.

Additional comments

The paper is improved.

Reviewer 3 ·

Basic reporting

OK, Please contineu to next process.

Experimental design

OK, Please contineu to next process.

Validity of the findings

OK

Additional comments

OK, Please contineu to next process.